# Organic Functionalized Graphene Oxide Behavior in Water

**DOI:** 10.3390/nano10061228

**Published:** 2020-06-24

**Authors:** Changwoo Kim, Junseok Lee, Will Wang, John Fortner

**Affiliations:** Department of Chemical and Environmental Engineering, Yale University, New Haven, CT 06520, USA; junseok.lee@yale.edu (J.L.); will.wang@yale.edu (W.W.)

**Keywords:** graphene oxide (GO), organic functionalization, colloidal stability, critical coagulation concentration (CCC)

## Abstract

Surface modified graphene oxide (GO) has received broad interest as a potential platform material for sensors, membranes, and sorbents, among other environmental applications. However, compared to parent (unmodified) GO, there is a dearth of information regarding the behavior of subsequently (secondary) modified GO, other than bulk natural organic matter (NOM) coating(s). Here, we systematically explore the critical role of organic functionalization with respect to GO stability in water. Specifically, we synthesized a matrix of GO-based materials considering a carefully chosen range of bound organic molecules (hydrophobic coatings: propylamine, tert-octylamine, and 1-adamantylamine; hydrophilic coatings: 3-amino-1-propanol and 3-amino-1-adamantanol), so that chemical structures and functional groups could be directly compared. GO (without organic functionalization) with varying oxidation extent(s) was also included for comparison. The material matrix was evaluated for aqueous stability by comparing critical coagulation concentration (CCC) as a function of varied ionic strength and type (NaCl, CaCl_2_, MgCl_2_, and MgSO_4_) at pH 7.0. Without surface derivatization (i.e., pristine GO), increased stability was observed with an increase in the GO oxidation state, which is supported by plate–plate Derjaguin, Landau, Verwey and Overbeek (DLVO) energy interaction analyses. For derivatized GO, we observed that hydrophilic additions (phi-GO) are relatively more stable than hydrophobic organic coated GO (pho-GO). We further explored this by altering a single OH group in the adamantane-x structure (3-amino-1-adamantanol vs. 1-adamantylamine). As expected, Ca^2+^ and monovalent co-ions play an important role in the aggregation of highly oxidized GO (HGO) and phi-GO, while the effects of divalent cations and co-ions were less significant for pho-GO. Taken together, this work provides new insight into the intricate dynamics of GO-based material stability in water as it relates to surface functionalization (surface energies) and ionic conditions including type of co- and counter-ion, valence, and concentration.

## 1. Introduction

Graphene oxide (GO) has received considerable attention as a promising platform material in various fields, including membrane technology, energy storage, sensors, and advanced catalysis due to its excellent physicochemical characteristics (low wall friction, high chemical stability, and relative high mechanical strength) [1,2]. GO consists of a two-dimensional (2D) lattice of hexagonal rings with oxygen-containing functional groups, including hydroxyl, epoxy, carboxyl, and carbonyl groups [3]. The type and quantity of oxygen-containing functionalities play an important role in GO aqueous stability, through the electrostatic double layer (EDL) and steric repulsion forces [4,5,6]. Furthermore, it has been recently demonstrated that the deprotonation of carboxylic groups is crucial for the stability of GO in its water phase [5]. The colloidal stability of GO is an important key in water-based applications, and accurate lifecycle assessments, including toxicity, and potential environmental fate and transport [6,7].

In aquatic systems, a number of common variables have been observed to affect the colloidal stability of GO, including pH, natural organic matter (NOM), and ionic strength/type [5,6,8,9]. In general, GO is stable in water over a wide range of pH due to oxygen-containing functional groups with a wide range of acid dissociation coefficient values (4–10) [10]. Thus, the stability of GO is pH dependent due to the different ionization degrees of the acidic oxygen-containing functional groups [8,11]. With the protonation of carboxyl groups (at the edge of GO), GO becomes more stable (hydrophilic) at higher pH conditions [5]. NOM, ubiquitous in the natural system in the range from 0.1 to 20 mg/L [12], has been observed to sorb on the basal plane of GO by strong π–π interactions or hydrophobic interactions [13]. Abundant carboxyl and hydroxyl groups from adsorbed NOM provide excellent colloidal stability of GO via EDL repulsion [8,9]. With a typical broad range of carbon functionality, including acetal, aromatic, heteroaliphatic, and aliphatic carbons, adsorbed NOM can also provide steric repulsion, thus enhanced stability [9]. In addition, the EDL of GO is affected by ionic strength and salt type (counter-ion valance and hydrated radius) [5,6]. It is well documented that high(er) ionic strength suppresses the EDL of GO, resulting in a reduction of colloidal stability. Moreover, multivalent ions can significantly decrease the colloidal stability of GO via bridging and cross-linking behaviors [5]. With identical counter-ion valence, smaller hydrated counter-ions more effectively screen the surface charge of GO [6].

For charge-stabilized nanomaterials, the stability ratio (reciprocal value of attachment efficiency) provides fundamental insight into particle stability regimes [14,15,16]. This value is obtained by measuring critical coagulation concentrations (CCC), which is widely used as a practical index for evaluating and comparing the stability of nanomaterials in water. In the case of spherical nanomaterials (sphere–sphere geometries), the CCC ratio of divalent ions to monovalent ions is approximated as 2^−6^, which is described via the Schulze–Hardy rule [17]. Although GO is 2D (not a spherical) nanomaterial, previous studies reported that GO does basically follow the Schulze–Hardy model derived from spherical approaches [18,19]. It is likely that GO forms a sphere-like aggregates during aggregation and hydration processes [20]. For flat materials, the theoretical calculation for plate–plate geometries as they relate to a CCC relationship is 2^−2^ (as derived in this work). To date, organically (surface) functionalized GO materials have been described for a variety of applications, including sensing, selective affinity, and membrane fabrication [21,22,23]. However, the role of surface derivatization, with small organic molecules that can provide not only EDL repulsion but also extended Derjaguin, Landau, Verwey and Overbeek (DLVO) interactions, such as osmotic and elastic-steric repulsion, has not been systematically evaluated [24,25].

Here, we explore the role of GO organic surface derivatization with regard to the oxidation extent, structure, and coating hydrophilicity. To accomplish this, we designed surface derivatized GO with a series of organic molecules (propylamine, tert-octylamine, 1-adamantylamine, 3-amino-1-propanol, and 3-amino-1-adamantanol), and a pristine GO (with varied oxidation extent), for comparison. The synthesized GO materials were characterized by Fourier transform infrared spectroscopy (FTIR), X-ray photoelectron spectroscopy (XPS), X-ray diffraction (XRD), and dynamic light scattering (DLS). The colloidal stability of the GO materials was evaluated and compared by measuring the CCC in the varied salt conditions (NaCl, CaCl_2_, MgCl_2_, and MgSO_4_). Additionally, we compared the measured CCC and their theoretical interaction energies in terms of attachment efficiencies.

## 2. Materials and Methods

### 2.1. Chemicals

Sulfuric acid (H_2_SO_4_, 98%), nitric acid (HNO_3_, 90%), graphite (powder, <45 μm, 99.99%), potassium permanganate (KMnO_4_, 99%), hydrogen peroxide (H_2_O_2_, 30%), sodium hydroxide (NaOH), sodium bicarbonate (NaHCO_3_), propylamine (>99%), tert-octylamine (95%), 1-adamantylamine (97%), 3-amino-1-propanol (99%), and 3-amino-1-adamantanol (96%) were purchased from Sigma-Aldrich. All materials were used as received.

### 2.2. Synthesis of Graphene Oxide (GO)

GO was synthesized by the modified Hummers method [26,27] using graphite (1 g), KMnO_4_ (3 g), and H_2_SO_4_ (25 mL). To control the oxidation extent of GO, the mixtures were separately heated at 30 °C and 50 °C for 2 h [23]. After heat reaction, the mixture was cooled in an ice bath. The resulting mixture was diluted by 175 mL of deionized (DI) water (>18.2 MΩ cm resistivity, Milli-Q, Millipore Corp). Then, the mixture was reheated at 60 °C for 30 min. After heating, we added H_2_O_2_ to remove the residual permanganate. The resulting suspension was purified with DI water until the pH of washed solution became neutral. The purified suspension (graphite oxide power) was filtrated and dried at 60 °C for 24 h. The GO suspension was obtained by the probe-sonication of graphite oxide power (150 mg) in DI water (200 mL) for 2 h. After sonication, the suspension was centrifuged at 12,000 rpm for 2 h. The resulting supernatant (GO) was kept in a glass bottle covered with aluminum foil.

### 2.3. Functionalization of GO

The surface of GO was organically functionalized using propylamine, tert-octylamine, 1-adamantylamine, 3-amino-1-propanol, and 3-amino-1-adamantanol, respectively. We added 300 mL of GO suspension (500 mg/L) and 0.1 g of coating agent into a three neck flask and heated the solution at 60 °C for 4 h. After the reaction, the resulting solution was purified with DI water over six times using stirred cell (100K Dalton, Millipore, Burlington, MA, USA) at 1 bar (N_2_ gas).

### 2.4. Critical Coagulation Concentration (CCC)

The CCC of the GO was measured by dynamic light scattering (DLS, NanoBrook Omni, Brookhaven Instruments Co., Holtsville, NY, USA) at 22 °C using NaCl, CaCl_2_, MgCl_2_, and MgSO_4_. Detailed information for measuring CCC has been described in our previous research and by others [28,29,30].

### 2.5. Hydrodynamic Diameter and Zeta Potential

The hydrodynamic diameter and zeta potential of GO materials were measured by DLS (NanoBrook Omni, Brookhaven Instruments Co., Holtsville, NY, USA) at 22 °C in 1 mM NaCl as background condition.

### 2.6. Fourier Transform Infrared Spectroscopy (FTIR)

An FTIR spectrum was measured using Nicolet 6700 FT-IR spectrometer (Thermo Scientific, Waltham, MA, USA).

### 2.7. X-Ray Photoelectron Spectroscopy (XPS)

An XPS (PHI VersaProbe II, Physical Electronics Inc., Chanhassen, MN, USA) was used with a monochromatic Al Kα X-ray source with a 0.47 eV system resolution.

### 2.8. X-Ray Diffraction (XRD)

XRD patterns were obtained using a Rigaku SmartLab X-ray Diffractometer (Rigaku, Tokyo, JP) with a scan step of 10° min^−1^ operating at 40 kV and 30 mA using Cu-Kα radiation.

## 3. Theory

### 3.1. The Plate–Plate Geometry DLVO Interactions

Classical Derjaguin, Landau, Verwey and Overbeek (DLVO) theory was used to calculate the interaction energy between the GO. Interaction energy (per area) profiles between GO and GO were interpreted upon the plate–plate geometry. Equations (1) and (2) for calculating electrostatic (*V_el_*) and van der Waals (*V_vdw_*) potential energy are as follows, respectively [31,32]:(1)$$V_{el}=\frac{2\sigma^{2}}{\varepsilon \varepsilon_{0}\kappa }\exp(-\kappa d)$$
(2)$$V_{vdw}=-\frac{A_{H}}{12\pi }\left(\frac{1}{d^{2}}+\frac{1}{{\left(d+2t\right)}^{2}}-\frac{2}{{\left(d+t_{1}\right)}^{2}}\right)$$
where *σ* is the surface charge density, *ε* is the dielectric constant of water, *ε_0_* is the permittivity of vacuum, *κ* is the inverse of the Debye length, *d* is the interaction distance between GO, *A_H_* is the Hamaker constants (GO−water−GO system), which were 4.9 × 10^−20^ and 13.5 × 10^−20^ J for GO and highly oxidized GO (HGO), respectively [4], and *t* is the thickness of GO, which was 6.7 Å [33]. The Debye length is given by
(3)$$\kappa^{-1}=\sqrt{\frac{\varepsilon \varepsilon_{0}k_{B}T}{2N_{A}e^{2}I}}$$
where *k_B_* is the Boltzmann constant, *T* is the absolute temperature, *N_A_* is the Avogadro number, *e* is the elementary charge, and *I* is the ionic strength. The surface charge density is given by
(4)$$\sigma =\frac{\varepsilon \varepsilon_{0}\kappa k_{B}T}{e}\sinh\left(\frac{e\psi }{k_{B}T}\right)$$
where *ψ* is the surface potential, which is calculated by the following equation:(5)$$\psi =\frac{4k_{B}T}{e}\mathrm{arctanh}\left[\exp(\kappa d_{Stern})\tanh\left(\frac{e\xi }{4k_{B}T}\right)\right]$$
where *d_Stern_* is the thickness of the stern layer (0.7 Å) [4], and *ξ* is the zeta potential.

### 3.2. Schulze–Hardy Rule (CCC Relationship for Sphere–Sphere Interaction)

Critical coagulation concentration (CCC) relationships in terms of counter valence ion concentration are described by the Schulze–Hardy rule, which is derived from linear superposition of Gouy–Chapman and unretarded Hamaker expressions (Equations (6) and (7)) [14,16]:(6)$$V_{el}=\frac{64\pi an_{\infty }k_{B}T}{\kappa^{2}}\gamma^{2}\exp(-\kappa d)$$
(7)$$V_{vdw}=-\frac{A_{H}a}{12d}$$
where *a* is diameters of particles, *n_∞_* is the bulk number density of ions, and *γ* is the reduced surface potential, which is given by
(8)$$\gamma =\tanh\left(\frac{ze\psi }{4k_{B}T}\right)$$
where *z* is the valance.

At the CCC, the total energy interaction (*V_T_* = *V_el_* + *V_vdw_*) and the gradient d*V_T_*/d*d* are zero. In this point, *κd* should be 1. Therefore, the bulk number density of ions can be expressed by
(9)$$n_{\infty }=\frac{288\pi^{2}{\left(k_{B}T\right)}^{5}{\left(\varepsilon \varepsilon_{0}\right)}^{3}\tanh^{4}(ze\psi /4k_{B}T)}{A_{H}^{2}e^{4}z^{6}}$$

Simply, CCC is proportional to the following equation:(10)$$CCC\propto z^{-6}\tanh^{4}(ze\psi /4k_{B}T)$$

### 3.3. CCC Relationship for Plate–Plate Interaction

We derived the CCC relationship with respect to the valance of counter ions by classical plate–plate geometry DLVO interaction (Equations (1) and (2)). The total energy interaction and its gradient d*V_T_*/d*d* are zero at the CCC (2*κ* = *d*). Here, the surface potential is calculated by the following equation, which is valid at a low surface potential (*zeψ* < 2*k_B_T*):(11)$$\sigma =\varepsilon \varepsilon_{0}\kappa \psi$$

Calculated bulk number density of ions can be expressed by
(12)$$n_{\infty }=\frac{1000\cdot 48\cdot \pi \exp(-2)k_{B}T{\left(\varepsilon \varepsilon_{0}\right)}^{2}\psi^{2}}{A_{H}e^{2}z^{2}}$$

CCC is proportional to the following equation:(13)$$CCC\propto z^{-2}\psi^{2}$$

## 4. Results and Discussion

### 4.1. Synthesis and Characterization of GO Materials

GO and highly oxidized GO (HGO) were synthesized by the modified Hummers method [26,27], whereby the extent of oxidation was controlled by the reaction temperature of the second stage (detailed information is described in Materials and Methods) [23]. The synthesized GO was surface functionalized with a series of organic molecules, including propylamine, tert-octylamine, 1-adamantylamine, 3-amino-1-propanol, and 3-amino-1-adamantanol (termed here as propylamine GO, tert-octylamine GO, 1-adamantylamine GO, 3-amino-1-propanol GO, and 3-amino-1-adamantanol GO, respectively). All the synthesized GO materials were well dispersed in water. Synthesized GO materials were characterized using XPS, FTIR, XRD, and DLS. As shown in the XPS measurement (Figure 1 and Table 1), the C1S spectra of the GO materials is deconvoluted into five types of carbon groups, represented as C–C, C–O, C–O–C, C=O, and O–C=O with binding energies centered at 284.6, 285.6, 286.7, 288.2, and 289.4 eV, respectively, as described by others [34]. As presented in Figure 1a, HGO has a higher percentage of oxidized carbon when compared with GO. The relative portion of oxidized carbon was 58.25% and 54.58% for HGO and GO, respectively. XPS survey indicated that the C/O atomic ratio of HGO and GO was 1.99 and 2.37, respectively. Upon organic functionalization, we observed an increase in C–O/C–N bonds as the amine groups were used as a linker for functionalization (in all cases). FTIR spectra in Figure 2a identifies the successful organic functionalization of GO, in which a amine peak (1600 cm^−1^) appears upon the functionalization for all materials [35]. We also observed decrease in % transmittance of OH stretch as the amine groups formed covalent amide bonds with the carboxyl or hydroxyl group of GO, which is in good agreement with the XPS results [36]. As shown in Figure 2b, XRD peak shifts were observed with organic functionalization. Diffraction peak at 10.1° (0.87 nm of d-spacing) of a pristine GO was shift to 9.1° (0.97 nm), 8.8° (1.00 nm), and 8.7° (1.01 nm) for propylamine GO, 3-amino-1-propanol GO, and tert-octylamine GO, respectively. As reported by others, functionalization increases interstitial spacing, thus the thickness of GO materials [37]. With respect to the adamantane groups, 2θ response of 1-adamantylamine GO and 3-amino-1-adamantanol GO disappeared, suggesting that bulk d-spacing(s) were irregular and/or uneven, which is likely due to the steric effect(s) of adamantane groups.

Lateral nanosheet size (*L*) and the surface zeta potential of the GO materials were measured using DLS at pH 7.0. Though the GO materials are not spherical particles, lateral nanosheet size (*L*) of GO materials are empirically estimated by DLS; *L* = (0.07 ± 0.03) × *D_H_*^1.5±0.15^, where *D_H_* is the hydrodynamic diameter [38]. *L* of GO and HGO was 85.0 ± 4.0 and 83.4 ± 8.8 nm, respectively. L of organic functionalized GO was slightly increased; 98.3 ± 8.1, 89.3 ± 2.9, 89.4 ± 1.6, 88.0 ± 6.4, and 87.6 ± 5.2 nm for propylamine GO, tert-octylamine GO, 1-adamantylamine GO, 3-amino-1-propanol GO, and 3-amino-1-adamantanol GO, respectively. Similar material sizes indicate that GO functionalization occurs without aggregation and the resulting products are relatively stable. Zeta potential of GO and HGO was −33.1 ± 1.9 and −43.3 ± 1.7 mV, respectively. HGO showed large negative zeta potential compared with GO because HGO contained dense oxygen-containing functional groups. Hydrophobic organic coated GO (pho-GO) including propylamine GO, tert-octylamine GO, and 1-adamantylamine GO, have relatively small(er) negative zeta potential compared to GO (ca. 10 mV). Zeta potential of propylamine GO, tert-octylamine GO, and 1-adamantylamine GO was −22.9 ± 1.1, −26.3 ± 2.1, and −22.2 ± 0.9 mV, respectively. Again, this is attributed to the fact that GO effectively loses carboxyl groups during functionalization. In case of hydrophilic organic coated GO (phi-GO), zeta potential of 3-amino-1-propanol GO and 3-amino-1-adamantanol GO are similar to that of GO as the carboxyl groups of GO are replaced with hydroxyl groups of 3-amino-1-propanol and 3-amino-1-adamantanol. Zeta potential of 3-amino-1-propanol GO and 3-amino-1-adamantanol GO was −31.6 ± 0.8 and −31.4 ± 2.8 mV, respectively.

### 4.2. Effect of Oxidation Extent on Colloidal Stability of GO

The effect of the oxidation extent of GO on colloidal stability was evaluated using GO and highly oxidized GO (HGO). As shown in Figure 3, the CCC values were measured using NaCl, CaCl_2_, MgCl_2_, and MgSO_4_ at pH 7.0 ± 0.2, respectively (details are also summarized in Table 2). Under all the ionic conditions tested, HGO shows the higher colloidal stability compared to GO; the CCC values in NaCl, CaCl_2_, MgCl_2_, and MgSO_4_ were 246.38, 1.40, 3.97, and 4.75 mM for GO, and 387.18, 1.79, 4.75, and 5.47 mM for HGO, respectively. As reported by others, the C/O ratios were negatively correlated with colloidal stability for the crumpled GO and carbon nanotubes [39,40]. In addition, the EDL repulsion is increased as a function of surface zeta potential through the formation of effective primary energy barrier [31,32].

The Schulze–Hardy rule, which relates the valence of counter ion(s) to the colloidal stability, can be used to predict the CCC values under ideal conditions. When considering plate–plate geometries, the ratio of monovalent counter ions to divalent counter ions is 2^2^ (as derived above). In the case of sphere–sphere geometry (Schulze–Hardy rule), the ratio is in the range between 2^2^ and 2^6^, which depends on the zeta potential. For larger zeta potentials, the ratio of CCC is proportional to 2^6^ due to the high EDL repulsion, and it moves towards 2^2^ as the zeta potential gets smaller (more hydrophobic). The results obtained in this work show that the ratio of NaCl to CaCl_2_, MgCl_2_, and MgSO_4_ is 2^7.46^, 2^5.96^, and 2^5.70^ for GO, and 2^7.76^, 2^6.45^, and 2^6.15^ for HGO, respectively. These obtained ratios are close to the expectation of the Schulze–Hardy rule for sphere–sphere geometry. It is likely that 2D shaped GO sheet dynamics appears to be more like 3D sphere-like aggregates during the aggregation processes (hydration dynamics may also play a role) [20]. Among the divalent counter ions (Ca^2+^ and Mg^2+^) evaluated, Ca^2+^ is stronger in influence for both GO and HGO, as Ca^2+^ has a higher propensity to form cross-linking interactions than Mg^2+^ [41]. We also observed that MgCl_2_ is more effective than MgSO_4_, which means that co-ions also, albeit to a lower extent, influence colloidal stability. With the same valance of counter ions, CCC values are inversely proportional to the valance of co-ions, which is termed the inverse Schulze–Hardy rule [42]. Compared with divalent co-ions, monovalent co-ions are more readily located near the GO surface, effectively neutralizing the surface charge [43].

To compare the experimental measurements with theoretical energy interactions, we calculated the classical plate–plate DLVO interaction where the attachment efficiency (α) is ca. 0.3, 0.8, and 1.0. Such plate–plate DLVO expressions are restricted to symmetric electrolytes because of the simplification of the Taylor series expansion in the EDL equation [44,45,46]. Thus, we selected NaCl as background ions, which is a 1:1 symmetric electrolyte. Figure 4 shows the DLVO interaction energy profiles of GO and HGO with respect to attachment efficiency. Under favorable conditions (*α* = 1), there are no primary maximums for both GO and HGO interaction energies. For a high α condition (*α* ≈ 0.8), 1.66 × 10^−4^ and 4.48 × 10^−4^ mJ/m^2^ of primary maximum were obtained at 1.3 and 0.9 nm (distance) for GO and HGO, respectively. For a low *α* condition (*α* ≈ 0.3), the highest primary maximum was observed; 1.01 × 10^−3^ and 6.72 × 10^−3^ mJ/m^2^ at 1.0 and 0.6 nm for GO and HGO, respectively. While the GO formed 3D sphere-like aggregates during the aggregation, plate–plate DLVO energy interactions were in good agreement with the CCC experimental measurements.

### 4.3. Effect of Organic Coating on Colloidal Stability of GO

The effects of surface coating on colloidal stability were evaluated using GO with five types of surface coatings; hydrophobic coatings (propylamine, tert-octylamine, and 1-adamantylamine) and hydrophilic coatings (3-amino-1-propanol and 3-amino-1-adamantanol). To evaluate their colloidal stability, we measured the CCC of both hydrophobic organic coated GO (pho-GO) and hydrophilic organic coated GO (phi-GO) under varied salt conditions (NaCl, CaCl_2_, MgCl_2_, and MgSO_4_) at pH 7 ± 0.2 (summarized in Table 2). As presented in Figure 5, the colloidal stability of pho-GO was decreased, compared to the pristine GO. The CCC values in NaCl were 101.4, 104.6, and 118.3 mM for propylamine GO, tert-octylamine GO, and 1-adamantylamine GO, respectively. Typically for GO, carboxyl groups play a significant role in EDL and steric repulsion [5]. As observed in the characterization of GO materials (Figure 1 and Figure 2), pho-GO effectively lost that carboxyl functionality. We also observed that the divalent cation type (Ca^2+^ vs. Mg^2+^) did not significantly influence the stability of pho-GO. CCC values in CaCl_2_, MgCl_2_, and MgSO_4_ were 1.1, 1.2, and 1.4 mM; 1.3, 1.6, and 1.7 mM; 2.0, 2.2, and 2.5 mM for propylamine GO, tert-octylamine GO, and 1-adamantylamine GO, respectively. In the presence of divalent cations, the colloidal stability of GO decreases through surface charge screening, bridging, intercalating, and cross-linking processes with oxygen-containing functional groups, including carboxyl, carbonyl, and hydroxyl groups [47]. Due to the deficit of the oxygen-containing (surface exposed) functionalities of pho-GO, the influence of type of divalent counter ions on colloidal stability is less significant.

Figure 6 presents the colloidal stability of phi-GO (3-amino-1-propanol GO and 3-amino-1-adamantanol GO). Compared with the pho-GO, phi-GO was more colloidally stable as expected. CCC values in NaCl were 153.1 (51% increase) and 303.3 mM (156% increase) for 3-amino-1-propanol GO and 3-amino-1-adamantanol GO, respectively. The coatings of both pho-Go and phi-GO have an identical structure and composition, except for a single hydroxyl functional group of phi-GO. We speculate that hydroxyl groups in phi-GO not only provide strong EDL repulsion (larger zeta potential), but also effectively hinder the hydrophobic interactions. Interestingly, we observed that Ca^2+^ has a stronger propensity to decrease the stability of phi-GO, compared with the pho-GO. CCC values in CaCl_2_, MgCl_2_, and MgSO_4_ were 1.2, 2.3, and 2.9 mM; 1.7, 2.8, and 3.5 mM for 3-amino-1-propanol GO and 3-amino-1-adamantanol GO, respectively. In the presence of divalent counter ions, Ca^2+^ ions play an important role as they uniquely coordinate between oxygen functionalities [5,47]. Additionally, we found that monovalent co-ions are more effective than divalent co-ions when considering the aggregation of phi-GO, compared to pho-GO. The role of co-ions in aggregation is based on surface charge screening [42,43]. It is likely that phi-GO has larger charged surface area (due to additional hydroxyl groups) than the pho-GO, which offers co-ion sensitivity for phi-GO.

## 5. Conclusions

In this work, we systematically describe the critical role of organic functionality as it relates to the colloidal stability of derivatized GO in water. Matrix variables including the oxidation extent of surface, coating structure, and hydrophilicity, have significant influence on the aqueous stability of GO. Generally, a higher oxidation extent provides higher effective colloidal stability. Furthermore, GO materials with hydrophobic organic coatings (pho-GO) are much less sensitive to divalent counter ion type and concentration(s). The presence of hydrophilic coatings improves the colloidal stability of GO by providing (or reconstituting) steric and EDL repulsions. As co-ions are directly related to the surface charge neutralization, monovalent co-ions have a higher impact on for those GO materials having a larger number of oxygen-containing functional groups (HGO and phi-GO). Taken together, this work highlights the role of organic functionality as it governs the stability of GO in water. Future studies should explore the transport of similarly derivatized GO in an environmental system.

## Figures and Tables

**Figure 1 nanomaterials-10-01228-f001:**
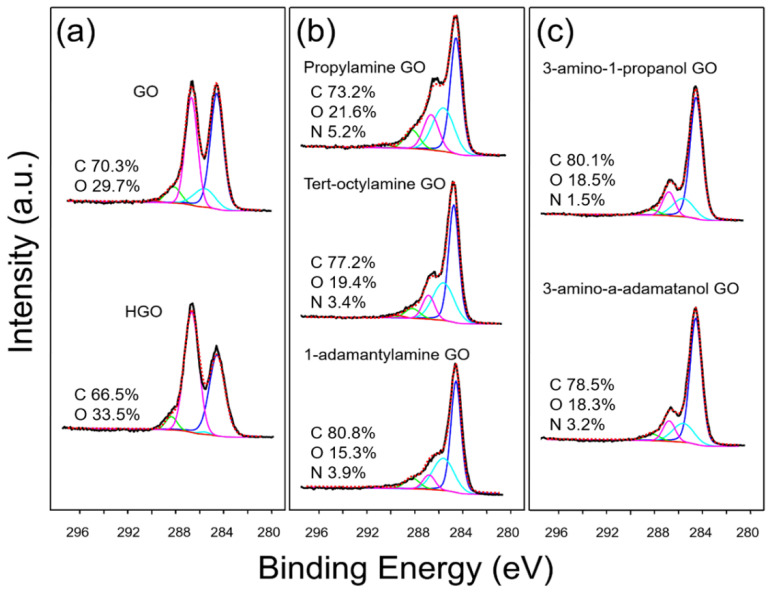
XPS spectra measurements (1S carbon (C1S) peak energies) of the GO materials; the C1S spectra is deconvoluted into five types, represented as C–C (blue), C–O (cyan), C–O–C (pink), C=O (green), and O–C=O (red) with the binding energies centered at 284.6, 285.6, 286.7, 288.2, and 289.4 eV, respectively. (**a**) GO and HGO (**b**) propylamine GO, tert-octylamine GO, and 1-adamantylamine GO (**c**) 3-amino-1-propanol GO and 3-amino-a-adamatanol GO. Relative atomic percentages were obtained by XPS survey spectra and included for each spectrum here.

**Figure 2 nanomaterials-10-01228-f002:**
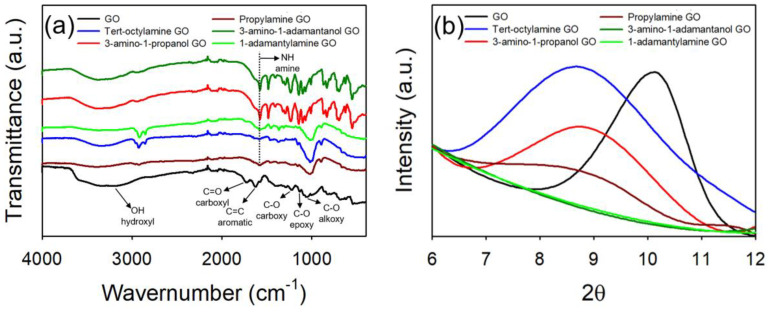
(**a**) FTIR spectra and (**b**) the diffraction patterns of GO (black), propylamine GO (dark red), tert-octylamine GO (blue), 1-adamantylamine GO (green), 3-amino-1-propanol GO (red), and 3-amino-a-adamatanol GO (dark green).

**Figure 3 nanomaterials-10-01228-f003:**
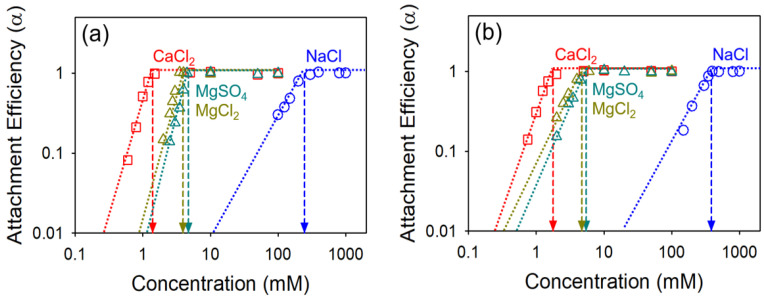
Attachment efficiency of (**a**) the GO and (**b**) the HGO as a function of salt concentration; NaCl (blue), CaCl_2_ (red), MgCl_2_ (dark yellow), and MgSO_4_ (dark green).

**Figure 4 nanomaterials-10-01228-f004:**
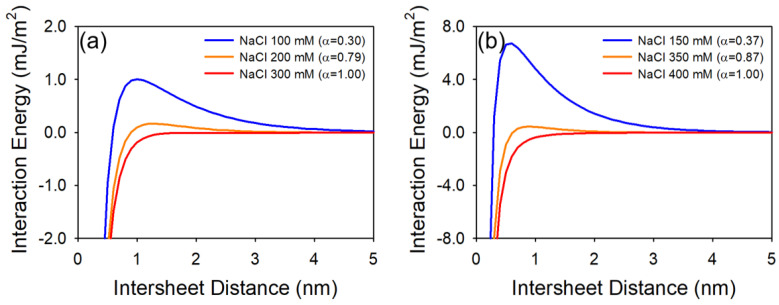
Derjaguin, Landau, Verwey and Overbeek (DLVO) interaction energy profiles (plate–plate geometry) of (**a**) GO and (**b**) HGO with respect to the attachment efficiency.

**Figure 5 nanomaterials-10-01228-f005:**
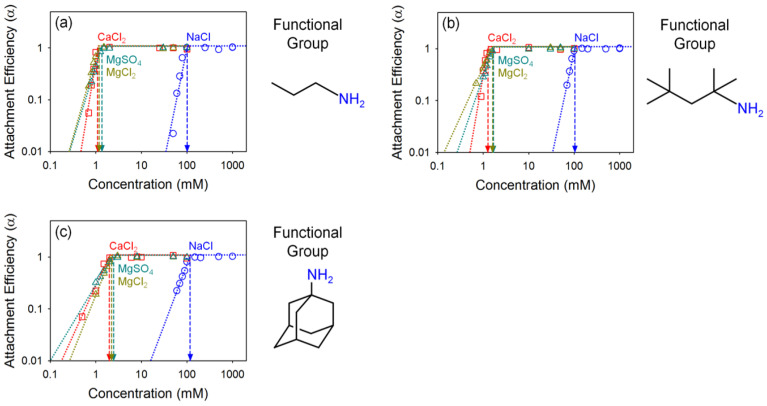
Attachment efficiency of (**a**) propylamine GO, (**b**) tert-octylamine GO, and (**c**) 1-adamantylamine GO as a function of salt concentration; NaCl (blue), CaCl_2_ (red), MgCl_2_ (dark yellow), and MgSO_4_ (dark green).

**Figure 6 nanomaterials-10-01228-f006:**
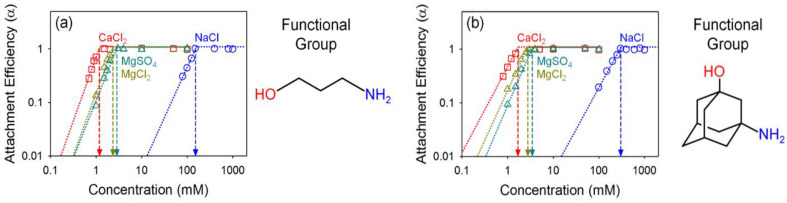
Attachment efficiency of (**a**) 3-amino-1-propanol GO, and (**b**) 3-amino-a-adamantanol GO as a function of salt concentration; NaCl (blue), CaCl_2_ (red), MgCl_2_ (dark yellow), and MgSO_4_ (dark green).

**Table 1 nanomaterials-10-01228-t001:** XPS binding energies of the individual peaks of the carbon 1S spectrum.

Sample	O-C=O (%)	C=O (%)	C-O-C (%)	C-O/C-N (%)	C-C (%)
Graphene oxide (GO)	0.55	6.62	36.53	10.89	45.42
Highly oxidized GO (HGO)	0.07	4.54	52.73	0.92	41.75
Propylamine GO	1.86	9.23	16.41	28.40	44.10
Tert-octylamine GO	1.38	5.56	11.15	30.65	51.27
1-adamantylamine GO	0.35	7.49	7.84	29.93	54.50
3-amino-1-propanol GO	0.00	3.40	12.41	16.45	67.74
3-amino-1-adamantanol GO	0.00	4.06	10.68	16.88	68.38

**Table 2 nanomaterials-10-01228-t002:** Summary of the critical coagulation concentration (CCC).

Samples	NaCl (mM)	CaCl_2_ (mM)	MgCl_2_ (mM)	MgSO_4_ (mM)	CCC Ratio				
					NaCl/ CaCl_2_	NaCl/ MgCl_2_	NaCl/ MgSO_4_	MgCl_2_/ CaCl_2_	MgSO_4_/ MgCl_2_
GO	246.38	1.40	3.97	4.75	2^7.46^	2^5.96^	2^5.70^	2.84	1.20
HGO	387.18	1.79	4.75	5.47	2^7.76^	2^6.45^	2^6.15^	2.65	1.15
Propylamine GO	101.38	1.10	1.18	1.37	2^6.42^	2^6.43^	2^6.21^	1.07	1.16
Tert-octylamine GO	104.55	1.28	1.62	1.69	2^6.46^	2^6.01^	2^5.95^	1.27	1.04
1-adamantylamine GO	118.29	1.99	2.23	2.49	2^5.90^	2^5.73^	2^5.57^	1.12	1.12
3-amino-1-propanol GO	153.05	1.19	2.34	2.88	2^7.01^	2^6.03^	2^5.73^	1.97	1.23
3-amino-1-adamantanol GO	303.31	1.70	2.79	3.51	2^7.48^	2^6.76^	2^6.43^	1.64	1.26
Schulze-Hardy Rule					2^2^–2^6^			1	
Plate-Plate Interaction					2^2^			1	

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
