# Peer review of "Organic Functionalized Graphene Oxide Behavior in Water"

_nanomaterials, 2020, doi:10.3390/nano10061228_

Round 1

Reviewer 1 Report

The manuscript presented for evaluation deals with graphene functionalized with compounds of interest, of organic nature. The work is carrefully performed and the conclusions are supported by the experimental part. A better discussion about the differences in the properties of GO and highly oxidized GO is recommended. As well, all abbreviation present in text should be explained. The manuscript can be accepted for publication after these very minor corrections. It was easier to follow the manuscript if a template is use with figures embedded in the main text. 

Reviewer 2 Report

The paper presents the critical role of organic functionality as it relates to colloidal stability of derivatized GO in water. The authors have discussed the colloidal stability of GO from multiple perspectives. The result shown in the paper is important for GO functionalization and its application. Therefore, I think that the paper would be acceptable for publication after minor revision.

1) Structural analysis using AFM is necessary. Do the derivatized GO exist as a single layer in water? How width and thick is each GO?

Reviewer 3 Report

This manuscript described the organic functionalized (hydrophilic or hydrophobic) graphene oxide behavior in water. The some results in this manuscript can be easily predicted without any experiment and some explanation for results is vague and not clear. I am just curious about which point is interesting and something new in this manuscript. 

  1. In Figure 1, some more details for XPS measurement can be needed.
  2. In line 217-219, Please explain the reason why the bulk d-spacing were irregular in the case of the adamantane groups.
  3. In line 224-226, I am just wondering whether L of organic functionalized GO is dependent on the functional groups or hydrophobicity.
  4. Please explain the difference of zeta potential between hydrophilic 1-propanol GO and 1-adamantanol GO.
  5. I think that the effect of organic coatings on colloidal stability can be easily predicted. What is important point and something new the author want to transfer to readers.   

Round 2

Reviewer 3 Report

The revised manuscript can be published as it is because the authors answered point by point about what I asked and commented.